# Effect of once-weekly dulaglutide on renal function in patients with chronic kidney disease

**Sungmin Kim**[1], **Jung Nam An**[1], **Young Rim Song**[1], **Sung Gyun Kim**[1], **Hyung Seok Lee**[1], **AJin Cho**[2]ʘ*, **Jwa-Kyung Kim**[1]ʘ*

**1** Department of Internal Medicine & Kidney Research Institute, Hallym University Sacred Heart Hospital, Anyang, Korea, **2** Department of Internal Medicine & Kidney Research Institute, Kangnam Sacred Heart Hospital, Seoul, Korea

ʘ These authors contributed equally to this work.
* kjk816@hallym.or.kr (JKK); ajinncho@gmail.com (AC)

## Abstract

### Background

Dulaglutide is associated with improved cardiovascular and kidney outcomes and can be a good therapeutic option for patients with type 2 diabetes with chronic kidney disease (CKD). In this study, the effects of dulaglutide on glucose-lowering efficacy and changes in renal function were analyzed.

### Methods

This retrospective study involved 197 patients with type 2 diabetes with mild-to-severe CKD treated with dulaglutide for at least 3 months between January 2017 and December 2020 at two tertiary hospitals in Korea. Changes in the creatinine-based estimated glomerular filtration rate (eGFR) and HbA1c were compared before and after the use of dulaglutide in each patient.

### Results

The number of patients and mean eGFR (mL/min/1.73 m$^2$) in CKD 2, 3a, 3b, and 4 were 94 (75.0 ± 8.5), 46 (54.8 ± 6.3), 31 (38.8 ± 4.4), and 26 (22.5 ± 5.4), respectively. Mean HbA1c level and body mass index (BMI) at the initiation of dulaglutide were 8.9% ± 1.4% and 29.1 ± 3.6 kg/m$^2$, the median duration of the use of dulaglutide was 16 months. The use of dulaglutide was associated with a mean decrease in HbA1c by 0.9% ± 1.5% and the glucose-lowering efficacy was similar across all stages of CKD. Also, it was associated with a reduced decline in the eGFR; the mean eGFR change after the use of dulaglutide was −0.76 mL/min/1.73 m$^2$ per year, whereas it was −2.41 mL/min/1.73 m$^2$ before use (paired *t*-test, *P* = 0.003). The difference was more pronounced in patients with an eGFR < 60 mL/min/1.73 m$^2$. Subgroup analysis showed that the renal protective effect was better in patients with proteinuria, age ≤ 65 years, and HbA1c < 9.0%, but showed no association with BMI.

**Data Availability Statement:** Data are available after the permission of the Ethics Committee for researchers who meet the criteria for access to confidential data. Hallym University Hospital

(Hallym University Institutional Review Board) prohibits any kind of patient information from releasing out of the hospital. If patient information is unavoidably provided externally, the patient information should be prevented from being identified. When looking at raw data, there is a potential risk even if all patient identification information has been deleted. So the data should be restricted. Data access requests may be directed to Miss Lim (contact via ssunnoa@gmail.com).

**Funding:** This research was supported by a National Research Foundation grant funded by the Korean government (2020R1A2C110138611).

**Competing interests:** The authors have declared that no competing interests exist.

## Conclusions

The use of dulaglutide provided adequate glycemic control irrespective of CKD stage and was associated with a reduced decline in the eGFR in the CKD population.

## Introduction

With the recent advances in diabetes treatment, customized prescriptions of antihyperglycemic medications tailored to an individual's risk have been available with different targets of glycated hemoglobin (HbA1c) [1–3]. However, the therapeutic options for patients with type 2 diabetes with chronic kidney disease (CKD) remain limited, especially in those with an estimated glomerular filtration rate (eGFR) <30–45 mL/min/1.73 m$^2$. In general, while recent guidelines advocate the use of sodium-glucose co-transporter 2 (SGLT2) inhibitors with metformin as first-line therapy, neither is recommended for initiation in advanced CKD [4–6]. In addition, in patients with very high HbA1c or persistent symptoms of hyperglycemia, the use of an SGLT2 inhibitor might increase symptoms of polyuria. In such cases, the next option for glycemic control is glucagon-like peptide-1 receptor agonists (GLP-1RAs) [7, 8].

GLP-1 is secreted by the L-cells of the small intestine in response to glucose ingestion [9]. Augmentation of GLP-1 results in improvement of beta-cell health in a glucose-dependent manner (post-prandial hyperglycemia) and suppression of glucagon (fasting hyperglycemia). As GLP-1 stimulates glucose-dependent insulin secretion, it can achieve significant reductions in HbA1c without the risk of hypoglycemia. In addition, GLP-1 analogs favor significant body weight (BW) loss and improvements in lipids and blood pressure (BP), therefore, they may be potentially beneficial, especially for obese diabetic CKD (DKD) patients. A high body mass index (BMI) is one of the most decisive risk factors for both new-onset CKD as well as the progression of pre-existing CKD in diabetes, and weight loss appears to be effective in reno-protection [10].

GLP-1RAs can be classified into two groups: incretin-mimetics and human GLP-1 analogues. Unlike the incretin-mimetics (exenatide, exenatide once-weekly, or lixisenatide), which have only 52% structural analogy to endogenous GLP-1 and thereby have great immunogenic power with potential development of inactivating antibodies, human GLP-1 analogues (liraglutide, dulaglutide, semaglutide, and albiglutide) have low immunogenic power due to a high structural analogy with endogenous GLP-1. Another fascinating aspect of the use of human GLP-1 analogues is that they can be initiated in patients with an eGFR as low as 15 mL/min/1.73 m$^2$. While most incretin-mimetics are 'short-acting' drugs, human GLP-1RAs are called 'long-acting' since specific molecular characteristics, such as the covalent bond with albumin (albiglutide), with the Fc portion of human immunoglobulin G4 (dulaglutide), or with specific fatty acids (liraglutide) give them a long half-life and prevent their elimination by the kidneys. Therefore, contrary to short-acting GLP-1RAs, human GLP-1 analogues induce a more marked reduction in HbA1c and fasting blood sugar and decrease the incidence of side effects [11, 12].

Based on the well-designed previous studies, the appropriate use of dulaglutide seems plausible in terms of and renal protection in DKD patients. However, DKD is a very heterogeneous disease, and the renal progression rate differs greatly depending on the clinical circumstances. So, a comparison of renal decline rate in each person before and after the use of dulaglutide may be needed to confirm the beneficial role of dulaglutide in renal function. Herein, with a retrospective review of medical records, we evaluated the efficacy of once-weekly dulaglutide in patients with mild-to-severe DKD and compared the changes in renal function before and after its use.

## Materials and methods

### Study design and participants

This retrospective study was performed in two tertiary hospitals in Korea. Patients with type 2 diabetes and mild-to-severe CKD who were treated with subcutaneous once-weekly dulaglutide for at least 3 months between January 2017 and December 2020 were included. All medical chart reviews were conducted after obtaining approval from the Hallym University Institutional Review Board (IRB No. 2022-04-004). All data were fully anonymized before we accessed them and the IRB waived the requirement for informed consent. Patients with stage 1 CKD were not included in this study because their renal function does not change rapidly. A total of 351 patients who were newly treated with dulaglutide were identified. Among these, 127 with stage 1 CKD and eight patients on dialysis were excluded. Nineteen patients who did not have previous blood data were also excluded. Finally, 197 patients were analyzed. All patients were treated with a standard dose of dulaglutide (0.75 or 1.5 mg) weekly. Concomitant use of other antihyperglycemic agents such as insulin, sulfonylurea, and metformin were also reviewed.

### Data collection

Baseline demographic, biochemical, and clinical data were obtained through chart reviews. To compare the changes in the eGFR before and after using dulaglutide directly in each patient, baseline serum creatinine (sCr) at the start of dulaglutide, as well as the previous sCr before using dulaglutide over a similar period, and sCr on the last date of the dulaglutide prescription were collected. The eGFR was calculated using sCr and the Chronic Kidney Disease Epidemiology Collaboration (CKD-EPI) equation. Proteinuria was assessed by using the urine albumin-to-creatinine ratio (UACR) or urine protein-to-creatinine ratio (UPCR) with spot urine samples. Microalbuminuria was defined as a UACR 30–300 mg/g. Patients with a UACR >300mg/g or UPCR >0.2 g/g were regarded as having proteinuria. Renal events were defined as (1) a doubling of serum creatinine, (2) a decrease in eGFR over 30%, or (3) progression to end-stage renal disease (ESRD) during the use of dulaglutide. Safety was also assessed through a chart review. Cases in which there was no mention of side effects were regarded as having no safety issues.

### Statistical analysis

Continuous variables were displayed as mean ± standard deviation (SD), and categorical variables as the number and proportion of patients. Demographic and clinical characteristics were described through absolute and relative frequencies (%), means, and/or SDs where appropriate. Changes in the eGFR from baseline to the last prescription date were calculated for each patient and compared with the changes in the eGFR before using dulaglutide. A paired *t*-test and the chi-squared test were used to compare the changes in the eGFR before and after using dulaglutide in each patient. Pearson correlation analyses were conducted to determine associations between changes in the eGFR and clinical parameters. A *P*-value < 0.05 was considered statistically significant. All statistical analyses were performed using SPSS 22.0 and GraphPad PRISM 9.

## Results

### Baseline characteristics

Of the 197 patients, 94 (47.7%), 46 (23.4%), 31 (15.7%), and 26 (13.2%) were in CKD stage 2, 3a, 3b and 4, respectively, and the mean eGFR was 75.0 ± 8.5, 54.8 ± 6.3, 38.8 ± 4.4, and

22.5 ± 5.4 mL/min/1.73 m$^2$, respectively. As shown in Table 1, the mean duration of diabetes was 14.5 years, and 72.6% of patients had a duration of over 10 years. The mean HbA1c level and BMI at the initiation of dulaglutide were 8.9% ± 1.4% and 29.1 ± 3.6 kg/m$^2$, respectively. One third of the patients had a BMI > 30 kg/m$^2$, and 19 (9.6%) had a BMI > 35 kg/m$^2$, a severe obesity. Concomitant antihyperglycemic medications were as follows. Fifty-nine patients (30.0%) received insulin, 131 (66.5%) received sulfonylurea, and most patients received metformin (81.2%). At baseline, urine data were available for 184 patients, among whom, 26.6%

**Table 1. Baseline characteristics.**

| Variables | Total (N = 197) | Changes in HbA1c (ΔHbA1c) | | |
|---|---|---|---|---|
| | | ≥ 0.9% (n = 97) | < 0.9% (n = 100) | p |
| Age (years) | 58.5 ± 12.8 | 58.3 ± 12.8 | 58.6 ± 12.8 | 0.876 |
| Sex, male, n (%) | 125 (55.8) | 59 (60.8) | 48 (48.0) | 0.048 |
| Duration of diabetes (years) | 14.5 ± 7.9 | 13.7 ± 8.1 | 15.4 ± 7.3 | 0.138 |
| > 10 years, n (%) | 143 (72.6) | 70 (72.2) | 73 (73.0) | 0.977 |
| DM retinopathy, n (%) | 102 (51.7) | 45 (46.3) | 57 (57.0) | 0.126 |
| HbA1c (%) | 8.9 ± 1.4 | 9.3 ± 1.3 | 8.3 ± 1.3 | <0.001 |
| > 9.0% n (%) | 87 (44.2) | 60 (61.9) | 27 (27.0) | <0.001 |
| SBP (mmHg) | 126.6 ± 30.2 | 126.5 ± 22.8 | 124.4 ± 29.6 | 0.220 |
| BMI (kg/m$^2$) | 29.1 ± 3.6 | 27.8 ± 4.0 | 28.7 ± 5.1 | 0.264 |
| > 30.0 n (%) | 61 (30.9) | 25 (25.8) | 36 (36.0) | 0.100 |
| > 35.0 n (%) | 19 (9.6) | 5 (5.1) | 14 (14.0) | 0.019 |
| Comorbidities, n (%) | | | | |
| Heart failure | 11 (5.6) | 7 (7.2) | 4 (4.0) | 0.213 |
| Coronary artery disease | 32 (16.2) | 13 (13.4) | 18 (18.0) | 0.319 |
| Cerebrovascular disease | 13 (6.6) | 4 (5.1) | 9 (9.0) | 0.174 |
| Lipid profiles | | | | |
| Total cholesterol | 157.4 ± 39.6 | 157.6 ± 37.9 | 157.1 ± 40.4 | 0.923 |
| LDL-cholesterol | 83.5 ± 31.0 | 82.1 ± 31.4 | 84.8 ± 30.7 | 0.550 |
| HDL-cholesterol | 46.6 ± 11.6 | 45.9 ± 11.3 | 47.2 ± 12.0 | 0.421 |
| Triglyceride | 194.0 ± 100.2 | 213.0 ± 107.4 | 177.5 ± 98.2 | 0.060 |
| Combined medication | | | | |
| Insulin | 59 (30.0) | 29 (29.9) | 30 (30.0) | 0.550 |
| Metformin | 160 (81.2) | 89 (91.8) | 81 (81.0) | 0.023 |
| Sulfonylurea | 131 (66.5) | 63 (64.9) | 68 (68.0) | 0.381 |
| RAS blocker | 140 (71.1) | 62 (63.9) | 78 (78.0) | 0.021 |
| Diuretics | 21 (10.7) | 8 (8.2) | 13 (13.0) | 0.250 |
| eGFR by creatinine (CKD-EPI) | | | | 0.990 |
| Baseline eGFR ≥60 to <90 | 94 (47.7) | 46 (47.4) | 48 (48.0) | |
| Baseline eGFR ≥45 to <60 | 46 (23.4) | 20 (20.6) | 26 (26.0) | |
| Baseline eGFR ≥30 to <45 | 31 (15.7) | 19 (19.6) | 12 (12.0) | |
| Baseline eGFR ≥15 to <30 | 26 (13.2) | 12 (12.4) | 14 (14.0) | |
| UACR (mg/g)* | | | | 0.333 |
| none (<30) | 77 (41.8) | 38 (41.8) | 39 (42.9) | |
| microalbuminuria (30–300) | 49 (26.6) | 21 (26.6) | 28 (30.8) | |
| proteinuria (>300) | 58 (31.5) | 34 (36.6) | 24 (26.4) | |

All data are expressed as mean ± SD,

* urine data was available in 184 patients

had microalbuminuria and 31.5% had proteinuria (UACR >300 mg/g or UPCR >0.2 g/g). The mean UACR in patients with microalbuminuria was 54.1 mg/g, and the UACR and UPCR of patients with proteinuria were 1011.6 mg/g and 1.9 g/g, respectively.

## Effects of dulaglutide on glucose control

The median duration of dulaglutide use was 16 (8–34) months, and it led to a mean decrease in HbA1c by –0.9 ± 1.5% ($P < 0.001$). The pre-, baseline, and post-use HbA1c levels are shown in Fig 1. The proportion of patients who achieved the HbA1c target of ≤7.0% and ≤6.5% was 27.0 and 18.8%, respectively. The glucose-lowering effect was better in males ($P = 0.048$) and those with higher baseline HbA1c levels ($P < 0.001$). However, the HbA1c-lowering effects were similar across all stages of CKD, and there was no difference in baseline renal function and the prevalence of proteinuria between patients with an HbA1c decrease ≥ –0.9% and < – 0.9% with dulaglutide (Table 1). Interestingly, the HbA1c-lowering effect was reduced in patients with severe obesity (BMI > 35 kg/m$^2$), although the baseline BMI was similar between the two groups (Table 1); in these patients with severe obesity, the HbA1c decreased by only – 0.48%. Correlation analysis showed that changes in HbA1c were closely related to sex (r = 0.141, $P = 0.049$), baseline HbA1c (r = –0.517, $P < 0.001$), diastolic BP (r = –0.180, $P = 0.030$), and baseline triglyceride level (r = –0.170. $P = 0.020$).

## Effects of dulaglutide on changes in renal function and proteinuria development

The change in eGFR *before* using dulaglutide was calculated with the blood test results from an average of 20 months before the baseline test. The change in eGFR *after* using dulaglutide was

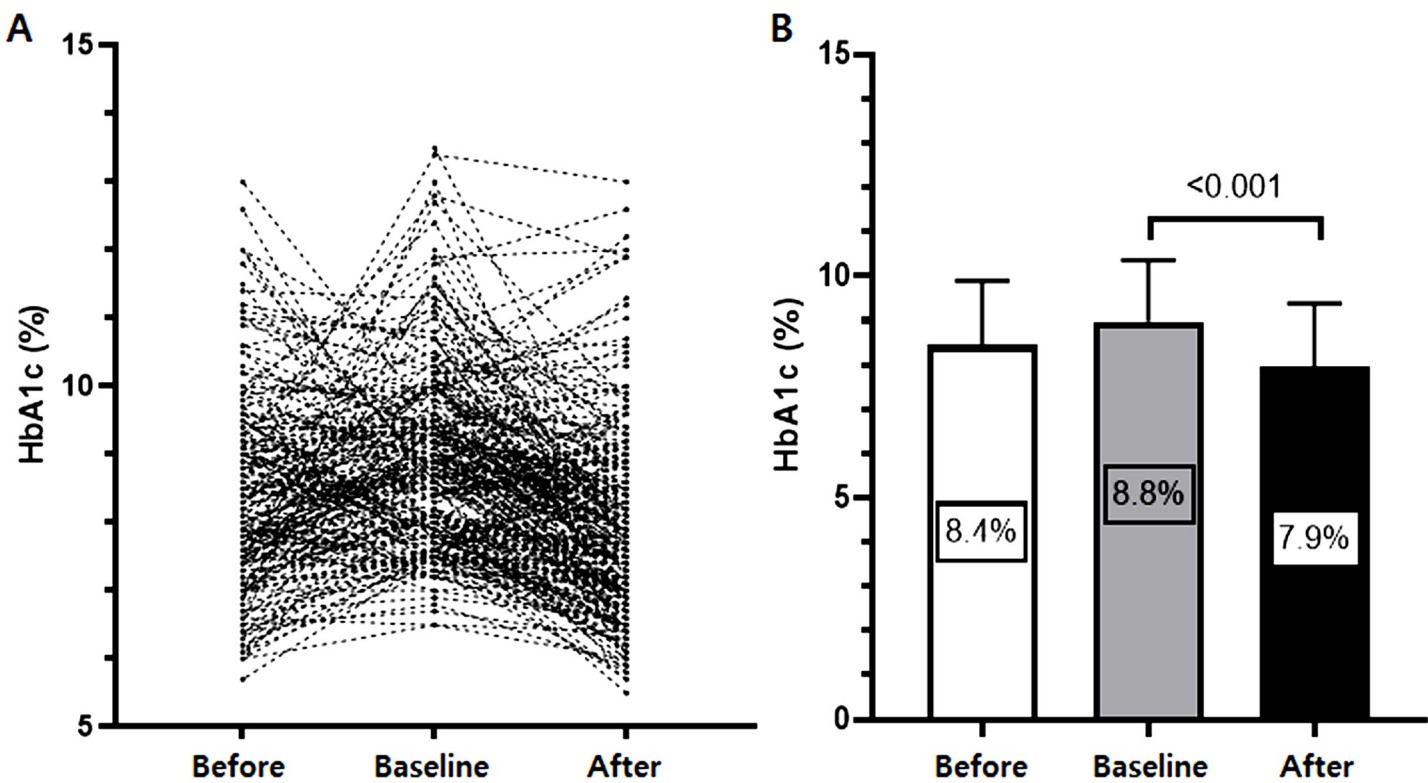

**Fig 1. Glucose-lowering efficacy.** (A) HbA1c levels at pre-, baseline, and post-dulaglutide use. (B) Once-weekly dulaglutide use significantly reduced HbA1c level.

calculated based on a blood test at approximately 15 months after the baseline test. Surprisingly, we found that the mean eGFR change before the use of dulaglutide was –2.41 mL/min/ 1.73 m² per year, but after the use of dulaglutide, the change decreased to –0.76 mL/min/1.73 m² per year. The difference was statistically significant (paired *t*-test, *P* = 0.003) (Fig 2A and 2B). In addition, the reduced decline in eGFR was more clearly observed in patients with moderate-to-severe CKD (Fig 2C): in patients with an eGFR < 60 mL/min/1.73 m², the change in the eGFR before and after the use of dulaglutide was –3.7 vs. –1.1 mL/min/1.73 m² per year (paired *t*-test, *P* < 0.001). However, in patients with an eGFR > 60 mL/min/1.73 m², there was no significant difference with the use of dulaglutide (–0.9 vs. –0.3 mL/min/1.73 m² per year, *P* = ns). In patients with CKD with an eGFR < 30 mL/min/1.73 m², the difference was marginally significant, probably because of the small number of patients (Fig 2D).

Also, the changes of eGFR before and after the use of dulaglutide were dependent on baseline proteinuria (Fig 3A). A significantly less decline in eGFR was observed in participants with baseline proteinuria compared with those without. The pre-and post- changes in the eGFR were –4.1 vs. –1.6 mL/min/1.73 m² per year in the proteinuria group (*P* = 0.008) and – 1.8 vs –0.4 mL/min/1.73 m² per year in the non-proteinuric group (*P* = 0.047). Additionally, we found that the effect of dulaglutide use on eGFR changes was better in relatively young patients (age ≤ 65 years) and in those with HbA1c ≤ 9.0%, suggesting that patients with very high HbA1c levels may have less renal benefit regardless of the dulaglutide glucose-lowering effect (Fig 3B and 3C). Also, it was noteworthy that the significant reduction in eGFR changes with dulaglutide use were consistently observed regardless of baseline BMI (Fig 3D).

For evaluation of the incidence of proteinuria development, we reviewed the urinalysis of 20 months before-, baseline, and 15 months after-dulaglutide use. A total of 145 cases and 184 cases were available pre- and post-dulaglutide use. As shown in Fig 4, the use of dulaglutide significantly reduced the development of proteinuria: before the use of dulaglutide, the progression of microalbuminuria to proteinuria occurred in 35.7% of the cases, but after the use of the dulaglutide, the incidence of proteinuria development from microalbuminuria occurred only in 7.0%. Whereas we did not find any difference in the incidence of new-onset microalbuminuria with the use of dulaglutide use.

During the use of dulaglutide, 11 patients (5.5%) experienced renal events. The mean baseline eGFR and the change in the eGFR of these patients were 35.0 ± 18.7 mL/min/1.73 m² and –9.2 mL/min/1.73 m² per year, respectively. It was significantly faster than the rate of patients

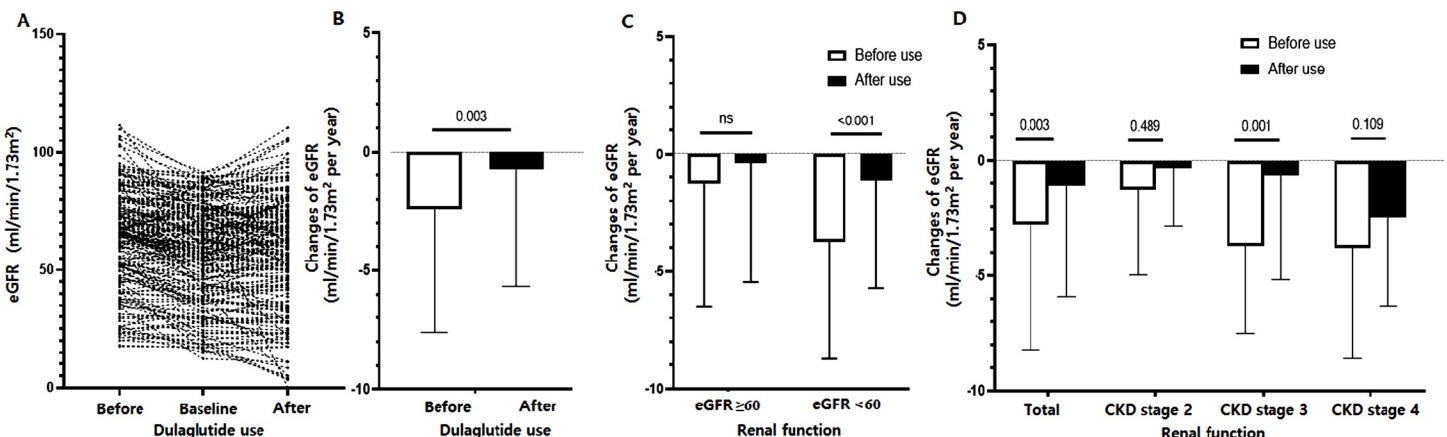

**Fig 2. eGFR values and mean eGFR change after the use of dulaglutide.** (A, B) The use of dulaglutide was associated with a significantly less decline of eGFR (paired t-test, P = 0.003). (C, D) The reduced decline in eGFR was more clearly observed in patients with moderate-to-severe CKD.

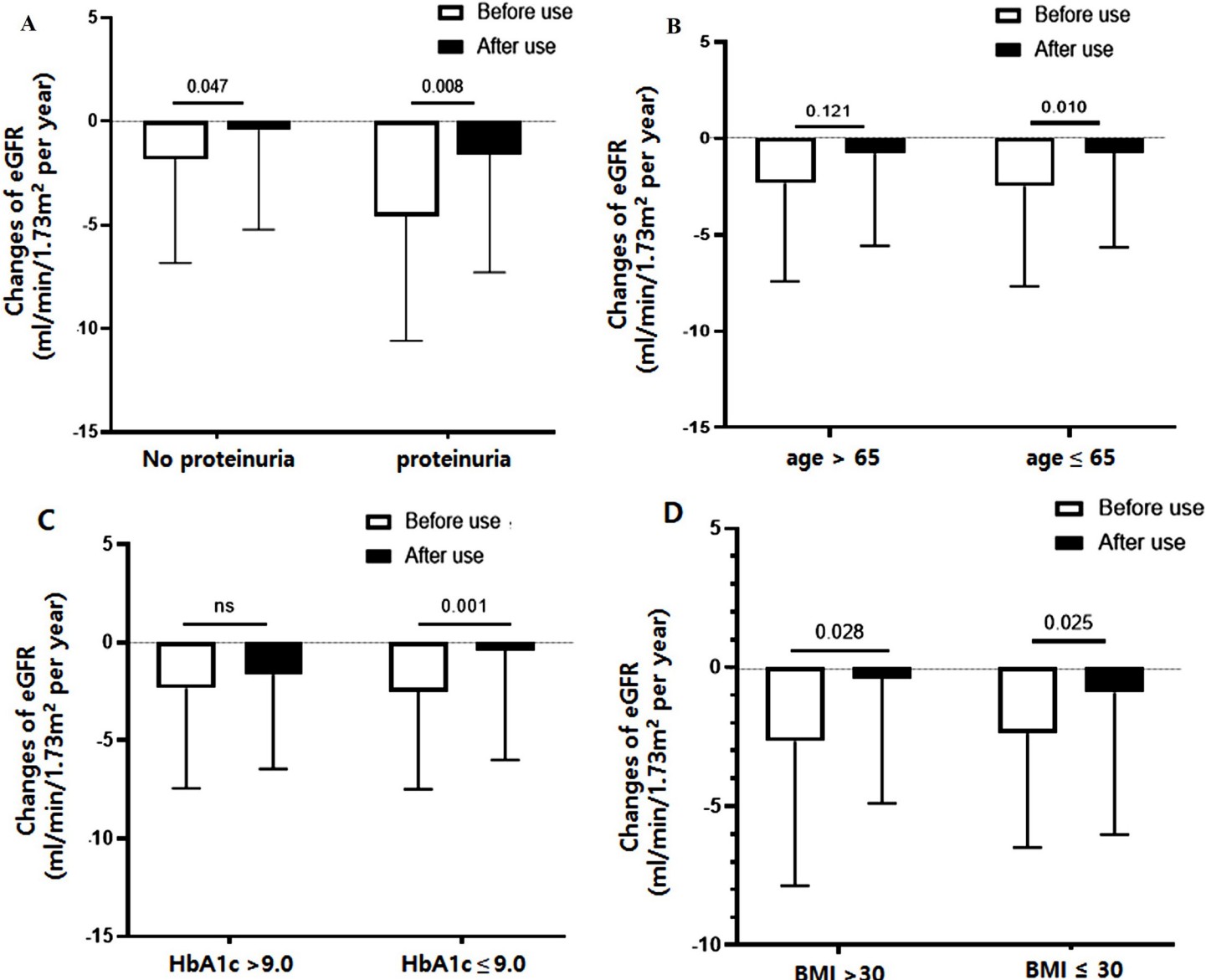

**Fig 3. Clinical factors affecting renal response to dulaglutide.** (A) A significantly less decline in eGFR was observed in participants with baseline proteinuria compared with those without. (B, C) The effect of dulaglutide use on eGFR changes was better in relatively young patients (age ≤ 65 years) and in those with HbA1c ≤ 9.0%. (D) However, changes in the eGFR were not associated with the baseline BMI.

without renal events. Most patients (n = 9) had proteinuria at baseline ($P < 0.001$), and significantly higher HbA1c (10.0 vs. 8.7%, $P = 0.002$) compared with those without renal events. Otherwise, age, BMI, BP, the duration of dulaglutide use, and the duration of diabetes were all comparable between the two groups.

## Adverse effects

Of the 197 patients, side effects were reported in 63 (31.9%), but discontinuation due to side effects was only seen in 10 cases (5.1%). Most of the side effects were gastrointestinal: diarrhea in 20 (10.1%), nausea in 34 (17.2%), and dyspepsia in 15 (8.7%). Injection site pain or allergic reaction was reported in only two cases. CKD stage showed no association with side effects.

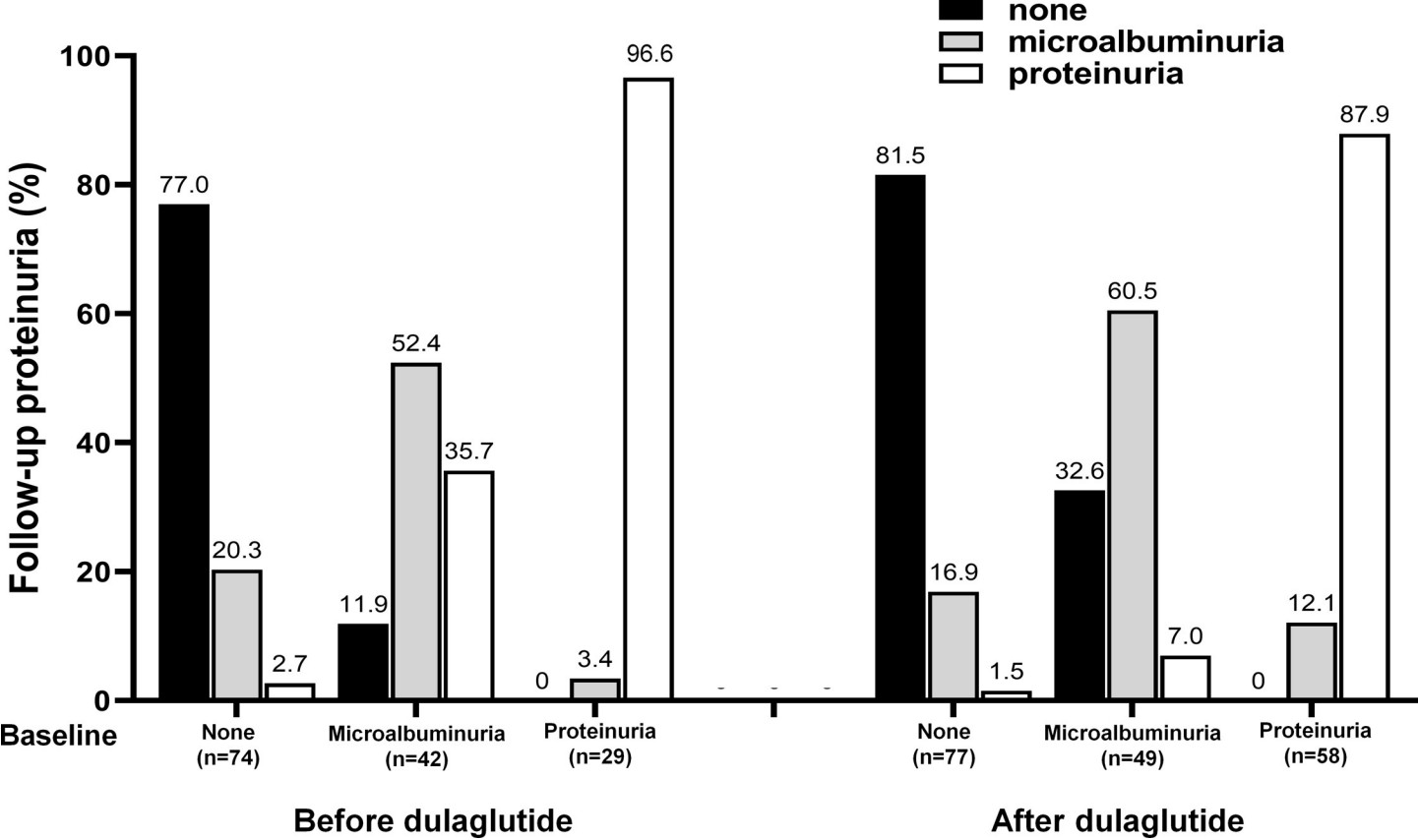

**Fig 4. The incidence rate of progression from microalbuminuria to proteinuria with dulaglutide use.** The rate of proteinuria development from microalbuminuria was significantly reduced after the use of dulaglutide. Urinalysis data were available in 145 and 184 cases before and after the use of dulaglutide, respectively.

## Discussion

In this study, we evaluated the glucose-lowering efficacy and renal effect of once-weekly dulaglutide therapy in patients with type 2 diabetes with mild-to-severe CKD in a real-world setting. Real-world experience shows that dulaglutide treatment for a median duration of 16 months decreases the mean HbA1c by 0.9%; the glucose-lowering effects were similar, irrespective of CKD stage. Also, we determined the effect of dulaglutide use on changes in renal function by comparing eGFR changes before and after the use of dulaglutide. Overall, the use of dulaglutide was associated with a significantly smaller decline in eGFR, and this was more clearly observed in patients with an eGFR $< 60$ mL/min/1.73 m$^2$ and in those with proteinuria. In addition, the renal response was weaker in older patients with critically high HbA1c ($> 9.0\%$), but BMI was not associated with the renal effect of dulaglutide.

CKD is one of the major complications for patients with diabetes. Previous studies have reported that 28%–43% of patients with type 2 diabetes have CKD, and the incidence keeps increasing despite advances in nephroprotective treatment [13]. In recent years, it has become apparent that diabetes-associated CKD is more heterogeneous than previously thought, so the term DKD is more commonly used to define CKD with diabetes than the traditional terminology of diabetic nephropathy which indicates histologically confirmed lesions [14]. In this regard, it is vital to optimize risk stratification and individualize therapeutic strategies based on each patient's clinical characteristics.

In moderate-to-severe CKD, GLP-1RAs can be a good therapeutic option [12, 15, 16]. A recently published systemic review and meta-analysis have shown that GLP-1RAs have beneficial effects on cardiovascular (CV) and renal outcomes beyond their blood glucose-lowering effects in patients with type 2 diabetes by reducing major adverse CV and renal events by 12% and 17%, respectively [17–21]. In particular, the mechanism of nephroprotective effects of GLP-1RAs is considered not only by the improvement of conventional risk factors for DKD, but also by direct actions on kidney including modulation of sodium and water homeostasis through sodium-hydrogen exchanger 3, improved tubule-glomerular feedback, reduced inflammation, and decreased oxidative stress [22, 23].

In this study, we summarized the experience of once-weekly dulaglutide for a median duration of 16 months in patients with DKD. First, we found that the HbA1c-lowering efficacy of dulaglutide was similar across CKD stages: −0.9%, −0.7%, −1.1%, and −0.9% in CKD stage 2, 3a, 3b, and 4, respectively. These findings are consistent with the outcomes of previous trials with dulaglutide. In the AWARD-2 trial, which compared the efficacy of dulaglutide and insulin glargine in patients with type 2 diabetes taking metformin and glimepiride, the use of dulaglutide at 1.5 mg/week produced a 1.0% reduction in HbA1c after 52 weeks. In the AWARD-7 trial, which included patients with moderate-to-severe CKD, HbA1c decreased by an average of 1.1% at 52 weeks [24]. Additionally, we found that the glucose-lowering efficacy was lower in severely obese patients, suggesting an insufficient response or resistance to dulaglutide therapy in severe obesity.

Next, we determined the effect of dulaglutide treatment on eGFR changes. For renal outcomes, two representative Randomized Clinical Trials (RCTs) have been conducted: the AWARD-7 trial and the Researching Cardiovascular Events with a Weekly Incretin in Diabetes (REWIND) trial. In the AWARD-7 trial, which included patients with moderate-to-severe CKD with a mean eGFR of 38 mL/min/1.73 m$^2$, dulaglutide treatment was associated with a smaller decline in eGFR compared with insulin glargine treatment [24]. Also in the REWIND trial, which included mostly low-risk patients with a mean eGFR of 75 mL/min/1.73 m$^2$, composite renal outcomes, mainly with the incidence of *de novo* proteinuria, occurred significantly less frequently in the dulaglutide group compared to placebo [25]. In addition, recently, the re-evaluation of the REWIND data with sensitivity analysis revealed that dulaglutide significantly reduces the worsening eGFR when it is defined as a reduction of ≥40% or ≥50%, rather than ≥30% in the original analysis, suggesting a determining role of dulaglutide in slowing the progression of renal damage [26]. Similarly, we also assessed eGFR changes associated with dulaglutide use in this study, too. As our study was not an RCT, we did not used non-users as control, but the changes in the eGFR before the use of dulaglutide were used as a control. Overall, the eGFR changes significantly decreased after the use of dulaglutide. Subgroup data showed that the renal effect was more pronounced in patients with an eGFR < 60 mL/min/1.73 m$^2$ or those with proteinuria. This observation is similar to the result of the AWARD-7 study, in which the reduced eGFR decline was most evident in participants with overt proteinuria. The reason for the greater absolute risk reduction in patients with increased albuminuria is that such patients have a higher absolute risk of developing a major kidney event. Furthermore, we found that after the dulaglutide use, the incidence of proteinuria. i.e, progression from microalbuminuria to proteinuria, was significantly reduced compared to the before-use of dulaglutide.

Lastly, the changes in the eGFR were not different according to baseline BMI. Considering that the glycemic-lowering effect of dulaglutide was weak in patients with severe obesity, there may be additional protective mechanisms of GLP-1RAs on renal benefit independent from glycemic control [27]. As mentioned above, these findings suggest that GLP-1RA may have a direct renal protective mechanism in addition to indirect effects such as conventional glucose

and BP control. Supporting our finding, previous reports also showed that there was no association between BW loss with GLP-1RA treatment and eGFR changes [28–30].

This study has several limitations. First, we used a retrospective chart review to compare renal function before and after the use of dulaglutide in patients with type 2 diabetes with mild-to-severe CKD. So, the effect of dulaglutide use on renal decline rate cannot be evaluated. And changes in renal function during the study period cannot be fully attributed to the drug. Second, as the data were not collected as a pre-designed requirement of the study, some urinalysis data were missing, Third, we could not obtain follow-up data on BW. It is well known that dulaglutide treatment is associated with significant weight loss. Changes in BW could cause a bias in eGFR changes with changes in muscle mass. However, results from AWARD-7 showed that cystatin-C-based eGFRs were consistent with creatinine-based eGFRs after dulaglutide administration. Furthermore, the effects of dulaglutide on renal function in this study were not different when measured by BMI [27]. Fourth, we cannot compare the does-response reno-protective effect by dulaglutide. In clinical practice, 0.75mg/week of dulaglutide is usually used during the first 2 weeks to monitor for side effects, after which the dose is increased to 1.5mg/week. So, in this study, we could not directly compare the efficacy between 0.75mg and 1.5mg. Finally, we obtained data on drug adverse events and causes of discontinuation retrospectively.

In conclusion, in patients with type 2 diabetes with CKD, dulaglutide use improved glycemic control irrespective of CKD stages. Additional benefits were also observed in reducing eGFR decline, particularly in patients with moderate CKD with proteinuria.

## Author Contributions

**Conceptualization:** AJin Cho, Jwa-Kyung Kim.

**Data curation:** Sungmin Kim, Jung Nam An, AJin Cho, Jwa-Kyung Kim.

**Formal analysis:** Sung Gyun Kim, Jwa-Kyung Kim.

**Investigation:** Jung Nam An.

**Methodology:** Hyung Seok Lee.

**Supervision:** Young Rim Song, Sung Gyun Kim.

**Validation:** Hyung Seok Lee, Jwa-Kyung Kim.

**Writing – original draft:** Sungmin Kim, AJin Cho, Jwa-Kyung Kim.

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
