## [Decision Letter · Decision Letter 0]

6 Jul 2022

PONE-D-22-17686Effect of once-weekly dulaglutide on renal function in patients with chronic kidney diseasePLOS ONE

Dear Prof. Kim,

Thank you for submitting your manuscript to PLOS ONE. After careful consideration, we feel that it has merit but does not fully meet PLOS ONE’s publication criteria as it currently stands. Therefore, we invite you to submit a revised version of the manuscript that addresses the points raised during the review process.

We look forward to receiving your revised manuscript.

Kind regards,

Tomislav Bulum

Academic Editor

PLOS ONE

Reviewers' comments:

Reviewer's Responses to Questions

**Comments to the Author**

1. Is the manuscript technically sound, and do the data support the conclusions?

Reviewer #1: Yes

Reviewer #2: Yes

Reviewer #3: Yes

2. Has the statistical analysis been performed appropriately and rigorously? 

Reviewer #1: No

Reviewer #2: Yes

Reviewer #3: Yes

3. Have the authors made all data underlying the findings in their manuscript fully available?

Reviewer #1: No

Reviewer #2: Yes

Reviewer #3: Yes

4. Is the manuscript presented in an intelligible fashion and written in standard English?

Reviewer #1: Yes

Reviewer #2: Yes

Reviewer #3: Yes

5. Review Comments to the Author

Reviewer #1: In this manuscript, the authors investigated renoprotective effects of dulaglutide in patients with type 2 diabetes (T2D) and CKD. They performed retrospective analysis and compared eGFR changes a year between before and after starting dulaglutide. They observed that dulaglutide use was associated with improvement of eGFR decline (-2.41 vs. -0.76 mL/min/1.73m2 per year). The renoprotective effects were observed strongly in patients with eGFR<60 (-3.7 vs. -1.1 mL/min/1.73m2 per year). Dulaglutide use resulted in reductions in HbA1c levels across the renal function.

Comments

1) Resolutions of Figures are not enough.

2) What was dose-response renoprotective effects by dulaglutide? Were any differences in 0.75 mg and 1.5 mg observed?

3) Changes in albuminuria by dulaglutide use should be shown.

4) I wonder if the effects of improvement of glycemic control on eGFR changes were excluded.

5) It seems that eGFR changes by dulaglutide use were robust compared to previous reports. What is specific reason underlying this observation?

6) Non-users of dulaglutide should be included as control.

Reviewer #2: The authors describe the effects of dulaglutide in a retrospective study of 197 type-2 diabetic patients with different CKD stages.

The paper is well written and the limitations of the study are clearly considered in the discussion.

I have no MAJOR CONCERNS.

- As MINOR COMMENTS i suggest the authors to consider some changes:

- The authors are using the term "macroalbumini¡uria!. Cam I suggest to change to that of "proteinuria " in files 191,193,203,229, 230, 231, 240, 241, 297, 312, 314 and 347 ?-

- Key words , file 74: GLP-1 receptor agonists

- file 151... weekly.

- file 303: Randomized Clinical Tral (RCT).

- file 326: body weight (BW).

-

-

Reviewer #3: Introduction

The introduction should be reorganized. In my opinion, comments on the results of clinical trials and meta-analysis (page 6 - lines 124-127) should be postponed for discussion. In this section, I would better describe the following aspects:

Page 5 – lines 104-105: “…while recent guidelines advocate the use of sodium-glucose co-transporter 2 105 (SGLT2) inhibitors with metformin as first-line therapy…”. I suggest inserting a bibliographic note in this sentence referring to the latest guidelines of the American Diabetes Association (available at the site https://diabetesjournals.org/care/issue/45/Supplement_1)

Page 5 – lines 117-119: regarding the safety profile and pharmacokinetics of GLP1-RAs, I believe it is necessary to make a brief clarification about the different pharmacological properties of incretion-mimetics and human GLP-1 analogues (see Granata et al. https:/doi.org/10.1093/ckj/sfac069).

Materials and methods

Page 7 - lines 162-164: please define more clearly the definitions of micro- and macroalbuminuria.

Discussion

Pag 13 lines - 294-301: it should be noted that the benefits of dulaglutide on worsening eGFR in the REWIND trial are underestimated. The re-evaluation of the data, conducted with the sensitivity analysis method, has shown that dulaglutide significantly reduces the worsening eGFR when it is defined as a reduction of ≥40% or ≥50%, rather than ≥30% as in the original study design (see Granata et al. https:/doi.org/10.1093/ckj/sfac069). Similar conclusions have been reported also in a recent meta-analysis (see Sattar et al. doi: 10.1016/S2213-8587(21)00203-5.), suggesting a determining role of GLP1-RAs in slowing the progression of renal damage beyond proteinuria. In my opinion, these aspects should be discussed in this section.

6. PLOS authors have the option to publish the peer review history of their article (what does this mean?). If published, this will include your full peer review and any attached files.

Reviewer #1: No

Reviewer #2: **Yes: **Alberto Martínez-Castelao

Reviewer #3: No

---

## [Author Response · Author response to Decision Letter 0]

23 Jul 2022

Dear Editor,

Dear Reviewers,

We greatly appreciate the opportunity to revise our paper in light of the reviewers’ comments and resubmit it for publication in PLOS ONE.

Manuscript Title: "Effect of once-weekly dulaglutide on renal function in patients with chronic kidney disease” 

Manuscript ID: PONE-D-22-17686

We made our best efforts in order to make the requested revisions in light of the editor and reviewer’s comments. Below you can find an itemized, point-by-point detailed response to all the questions and comments of the reviewers.

We hope our paper is now suitable for publication in PLOS ONE in its present form and we are now resubmitting it to your attention.

REVIEWERS’ Comments:

Reviewer #1 Comments:

In this manuscript, the authors investigated renoprotective effects of dulaglutide in patients with type 2 diabetes (T2D) and CKD. They performed retrospective analysis and compared eGFR changes a year between before and after starting dulaglutide. They observed that dulaglutide use was associated with improvement of eGFR decline (-2.41 vs. -0.76 mL/min/1.73m2 per year). The renoprotective effects were observed strongly in patients with eGFR<60 (-3.7 vs. -1.1 mL/min/1.73m2 per year). Dulaglutide use resulted in reductions in HbA1c levels across the renal function.

Comment:

1) Resolutions of Figures are not enough.

Answer:

In the revised manuscript, we increased the resolution of each Figure. Thank you.

Special Comment:

2) What was the dose-response reno-protective effects by dulaglutide? Were any differences in 0.75 mg and 1.5 mg observed?

Answer:

As you pointed out, it may be needed to compare the efficacy between 0.75mg and 1.5mg of dulaglutide. In clinical practice, however, 0.75mg/week of dulaglutide is usually used during the first 2 weeks to monitor for side effects, after which the dose is increased to 1.5mg/week. So, in this study, we could not directly compare the efficacy between 0.75mg and 1.5mg. But according to the AWARD-7, either dulaglutide 1.5mg or 0.75 mg was associated with a significantly smaller decline of eGFR from baseline compared with insulin glargine. We added this in the limitation of the revised manuscript. 

Comment:

3) Changes in albuminuria by dulaglutide use should be shown.

Answer:

Thank you for your comments. In the revised manuscript, we added the data comparing changes in urinary albumin before and after the use of dulaglutide (new Figure 4 in the revised manuscript). As shown in the figure below, the use of dulaglutide significantly reduced the incidence rate of progression from microalbuminuria to proteinuria. Before the use of dulaglutide, the progression of microalbuminuria to proteinuria occurred in 35.7% of the cases, but after the use of the dulaglutide, the incidence of proteinuria development from microalbuminuria occurred only in 7.0%. Whereas, we did not find any difference in the incidence of new-onset microalbuminuria with the use of dulaglutide use. We added this in the RESULT section of the revised manuscript. 

Comment:

4) I wonder if the effects of improvement of glycemic control on eGFR changes were excluded.

Answer:

Thank you for your kind comments. There was no correlation between HbA1c level after dulaglutide use and eGFR changes (r =0.006, p=0.938). However, as we compared the eGFR changes before and after the use of dulaglutide, we could not predict the eGFR slope with the use of dulaglutide. In this regard, we cannot precisely adjust the effect of glycemic control on the eGFR decline rate. But as shown in Figure 3C, the reduction in the eGFR decrease with the use of dulaglutide was not clear in patients with very high baseline HbA1c levels. We added this in the limitation section of the revised manuscript. 

Comment:

5) It seems that eGFR changes by dulaglutide use were robust compared to previous reports. What is specific reason underlying this observation?

Answer:

We totally agree with your opinion. 

We think that the clear difference observed in our study occurred because we compared the mean eGFR changes before and after the use of dulaglutide in each patient. As you know, diabetic kidney disease is a very heterogeneous disease entity, therefore, the renal progression rate differs greatly depending on the clinical circumstances. So, rather than analyzing all DKD patients with very diverse clinical manifestations, we compared before- and after-changes in each patient, so the result seems to be much more robust compared to other reports.

Comment:

6) Non-users of dulaglutide should be included as control.

Answer:

Thank you for your comments. As you mentioned, if this study evaluated the efficacy of dulaglutide in DKD patients, we definitely need non-users as control. But in this study, we compared the changes before and after the use of dulaglutide in each patient, so we did not include non-users. We added this in the DISCUSSION section in the revised manuscript. 

Reviewer #2 Comments:

General Comment:

The authors describe the effects of dulaglutide in a retrospective study of 197 type-2 diabetic patients with different CKD stages. The paper is well written and the limitations of the study are clearly considered in the discussion.

I have no MAJOR CONCERNS.

- As MINOR COMMENTS I suggest the authors to consider some changes:

- The authors are using the term "macroalbumini¡uria!. Cam I suggest to change to that of "proteinuria " in files 191,193,203,229, 230, 231, 240, 241, 297, 312, 314 and 347 ?-

- Key words , file 74: GLP-1 receptor agonists

- file 151... weekly.

- file 303: Randomized Clinical Tral (RCT).

- file 326: body weight (BW).

Answer:

Thank you for your kind comments. As you recommended, we all changed the “macroalbuminuria” to “proteinuria” in the revised manuscript. Also, all the typographical errors were corrected.

Reviewer #3 Comments:

Comment:

The introduction should be reorganized. In my opinion, comments on the results of clinical trials and meta-analysis (page 6 - lines 124-127) should be postponed for discussion. In this section, I would better describe the following aspects:

Page 5 – lines 104-105: “…while recent guidelines advocate the use of sodium-glucose co-transporter 2 105 (SGLT2) inhibitors with metformin as first-line therapy…”. I suggest inserting a bibliographic note in this sentence referring to the latest guidelines of the American Diabetes Association (available at the site https://diabetesjournals.org/care/issue/45/Supplement_1)

Page 5 – lines 117-119: regarding the safety profile and pharmacokinetics of GLP1-RAs, I believe it is necessary to make a brief clarification about the different pharmacological properties of incretion-mimetics and human GLP-1 analogues (see Granata et al. https:/doi.org/10.1093/ckj/sfac069).

Answer:

Thank you for your kind comments. As you recommended, reorganized the manuscript. The details about the results of RCTs and meta-analysis were shown in Discussion section in the revised manuscript. 

Instead, we added the pharmacological properties of incretin-mimetics and human GLP-1 RA in the Introduction section of the revised manuscript: “GLP-1RAs can be classified into two groups: incretin-mimetics and human GLP-1 analogues. Unlike the incretin-mimetics (exenatide, exenatide once-weekly, or lixisenatide), which have only 52% structural analogy to endogenous GLP-1 and thereby have great immunogenic power with potential development of inactivating antibodies, human GLP-1 analogues (liraglutide, dulaglutide, semaglutide, and albiglutide) have low immunogenic power due to a high structural analogy with endogenous GLP-1. Another fascinating aspect of the use of GLP-1RAs is that they can be initiated in patients with an eGFR as low as 15 mL/min/1.73 m2. While most incretin-mimetics are ‘short-acting’ drugs, human GLP-1RAs are called ‘long-acting’ since specific molecular characteristics, such as the covalent bond with albumin (albiglutide), with the Fc portion of human immunoglobulin G4 (dulaglutide), or with specific fatty acids (liraglutide) give them a long half-life and prevent their elimination by the kidneys. Therefore, contrary to short-acting GLP-1RAs, human GLP-1 analogues induce a more marked reduction in HbA1c and fasting blood sugar and decrease the incidence of side effects”. Also, we added the recent ADA guideline as new reference.

Comment:

Materials and methods

Page 7 - lines 162-164: please define more clearly the definitions of micro- and macroalbuminuria.

Answer:

To avoid confusion, the term “macroalbuminuria” was deleted from the revised manuscript and changed to “proteinuria”. Microalbuminuria was defined as a UACR 30–300 mg/g. Patients with a UACR >300mg/g or UPCR >0.2 g/g were regarded as having proteinuria.

Comment:

Discussion

Pag 13 lines - 294-301: it should be noted that the benefits of dulaglutide on worsening eGFR in the REWIND trial are underestimated. The re-evaluation of the data, conducted with the sensitivity analysis method, has shown that dulaglutide significantly reduces the worsening eGFR when it is defined as a reduction of ≥40% or ≥50%, rather than ≥30% as in the original study design (see Granata et al. https:/doi.org/10.1093/ckj/sfac069). Similar conclusions have been reported also in a recent meta-analysis (see Sattar et al. doi: 10.1016/S2213-8587(21)00203-5.), suggesting a determining role of GLP1-RAs in slowing the progression of renal damage beyond proteinuria. In my opinion, these aspects should be discussed in this section.

Answer:

Thank you very much for your kind comments. In the revised manuscript, we corrected the paragraph as follows. “Also in the REWIND trial, which included mostly low-risk patients with a mean eGFR of 75 mL/min/1.73 m2, composite renal outcomes, mainly with the incidence of de novo macroalbuminuria, occurred significantly less frequently in the dulaglutide group compared to placebo.[25] In addition, recently, the re-evaluation of the REWIND data with sensitivity analysis revealed that dulaglutide significantly reduces the worsening eGFR when it is defined as a reduction of ≥40% or ≥50%, rather than ≥30% in the original analysis.” Also, some more references were added.

Thank you so much for your kind comments.

Sincerely,

Jwa Kyung Kim, MD, PhD

Hallym University Sacred Heart Hospital, Korea

---

## [Editor Report · Decision Letter 1]

1 Aug 2022

Effect of once-weekly dulaglutide on renal function in patients with chronic kidney disease

PONE-D-22-17686R1

Dear Dr. Kim,

We’re pleased to inform you that your manuscript has been judged scientifically suitable for publication and will be formally accepted for publication once it meets all outstanding technical requirements.

Kind regards,

Tomislav Bulum

Academic Editor

PLOS ONE

Additional Editor Comments:

In revised manuscript term macroalbuminuria is still present in some part of Results and Discussion section. Please change to proteinuria.

---

## [Editor Report · Acceptance letter]

4 Aug 2022

PONE-D-22-17686R1 

Effect of once-weekly dulaglutide on renal function in patients with chronic kidney disease 

Dear Dr. Kim:

I'm pleased to inform you that your manuscript has been deemed suitable for publication in PLOS ONE. Congratulations! Your manuscript is now with our production department. 

Kind regards, 

on behalf of

Dr. Tomislav Bulum 

Academic Editor

PLOS ONE